# Abscisic Acid Enhances Trehalose Content via *OsTPP3* to Improve Salt Tolerance in Rice Seedlings

**DOI:** 10.3390/plants12142665

**Published:** 2023-07-17

**Authors:** Nenghui Ye, Yuxing Wang, Huihui Yu, Zhonge Qin, Jianhua Zhang, Meijuan Duan, Ling Liu

**Affiliations:** 1Hunan Provincial Key Laboratory of Rice Stress Biology, College of Agronomy, Hunan Agricultural University, Changsha 410128, China; laonengye@gmail.com (N.Y.); yuxingwangbae@gmail.com (Y.W.); huihuiyu2016@gmail.com (H.Y.); zhongeqin@gmail.com (Z.Q.); 2Key Laboratory of Crop Physiology and Molecular Biology, Ministry of Education, Hunan Agricultural University, Changsha 410128, China; 3Department of Biology, Hong Kong Baptist University, Kowloon, Hong Kong 999077, China; 4School of Life Sciences and State Key Laboratory of Agrobiotechnology, The Chinese University of Hong Kong, Shatin, Hong Kong 999077, China

**Keywords:** salt tolerance, trehalose, abscisic acid, *OsTPP3*, rice seedling

## Abstract

Salt stress is one of the major environmental stresses that imposes constraints to plant growth and production. Abscisic acid (ABA) has been well-proven to function as a central integrator in plant under salt stress, and trehalose (Tre) has emerged as an excellent osmolyte to induce salt tolerance. However, the interacting mechanism between ABA and Tre in rice seedlings under salt stress is still obscure. Here, we found that the application of exogenous Tre significantly promoted the salt tolerance of rice seedlings by enhancing the activities of antioxidant enzymes. In addition, the expression of *OsNCED3* was significantly induced by salt stress. The overexpression of the *OsNCED3* gene enhanced the salt tolerance, while the knockout of *OsNCED3* reduced the salt tolerance of the rice seedlings. Metabolite analysis revealed that the Tre content was increased in the *OsNCED3*-overexpressing seedlings and reduced in the *nced3* mutant. The application of both ABA and Tre improved the salt tolerance of the nced3 mutant when compared with the WT seedling. *OsTPP3* was found to be induced by both the ABA and salt treatments. Consistent with the *OsNCED3* gene, the overexpression of *OsTPP3* enhanced salt tolerance while the knockout of *OsTPP3* reduced the salt tolerance of the rice seedlings. In addition, the Tre content was also higher in the *OsTPP3*-overexpressing seedling and lower in the *tpp3* mutant seedling than the WT plant. The application of exogenous Tre also enhanced the salt tolerance of the *tpp3* mutant plant. Overall, our results demonstrate that salt-increased ABA activated the expression of *OsTPP3*, which resulted in elevated Tre content and thus an improvement in the salt tolerance of rice seedlings.

## 1. Introduction

Rice (*Oryza sativa* L.) is a staple food for half of the global population and comprises 23% of the calories consumed worldwide [1,2]. China is the largest rice producer and consumer in the world and feeds 20% of the global population with only approximately 5% of the planet’s water resources and 7% of its arable land [3]. Rapid population growth and economic development have given rise to a higher demand for food production [4,5]. However, it has been observed that rice is more sensitive to saline-alkali stress during the seedling stage (3-leaf stage) and reproductive stage (panicle initiation and fertilization) than that during seed germination and the final growth stages [6,7,8]. Furthermore, about one-third of land worldwide is threatened by saline-alkali stress, which is even further deteriorated by inappropriate field irrigation, industrial pollution, growing population, and soil salinization in coastal zones [9,10]. Saline-alkali stress is one of the major environmental stresses that has significantly inhibited plant growth and crop yield all over the world [11]. In China, more than a 100 million hectares of land area is saline or alkali affected, among which more than 6.7 million hectares of farmland that are technically suitable for rice production are left uncultivated or have very low yields due to saline stress [12]. Therefore, it is of great benefit to exploit and utilize saline-alkali land for agriculture production [13,14].

Salt stress can severely affect plant growth and development during the whole life cycle including seed germination, seedling establishment, and yield formation [15,16]. To survive from this harsh environmental stress, plants have evolved different response mechanisms to deal with salt stress such as salt tolerance, avoidance and escape, and the recovery mechanism [13,17]. The salt overly sensitive (SOS) pathway plays a key role in regulating cellular ion homeostasis during salt stress and represents the well-conserved salt tolerant mechanism in higher plants [5,7,18]. Salt elevates the cytosolic Ca^2+^, which activates the protein kinase complex, the SOS2-SOS3. The activated SOS2-SOS3 complex then activates SOS1, a plasma membrane Na^+^/H^+^ antiporter [19,20]. In rice, the *OsSOS1*, *OsSOS2*/*OsCIPK24*, and *OsSOS3*/*OsCBL4* genes have been isolated and the function and relationship between them have been well-revealed [18,21,22]. Except for ionic unbalance induced by salt stress, osmotic stress is another problem imposed on plants by salt stress. To cope with osmotic stress, plants thus accumulate compatible osmolytes such as sugar, proline, glycine betaine, polyamines, and proteins from the late embryogenesis abundant (LEA) superfamily [23,24], which play a dominant role in osmotic adjustment by reducing cell osmotic potential and stabilizing proteins and cellular structures [13,25].

Plant hormones are another crucial regulator for plant growth in response to salt stress, among which abscisic acid (ABA) functions as a central integrator for the developmental process and abiotic tolerance including salt tolerance [11,26]. ABA plays a major and irreplaceable role during salt stress response in plants by regulating ion homeostasis, the biosynthesis of osmolytes, reactive oxygen species (ROS) scavenging, and salt-responsive gene expression. As the main mediator of the plant stress response, ABA helps plants survive salt stress through integrating with the second messenger Ca^2+^ by provoking plasma membrane-bound channels or releasing Ca^2+^ from intracellular Ca^2+^ storage banks [27], indicating a crosstalk between the ABA signaling pathway and SOS pathway under salt stress. In response to salinity and osmotic stress, the ABA biosynthesis genes *OsNCED3* and *OsNCED5*, encoding 9-cis-epoxycarotenoid dioxygenases, increase the ABA levels and enhance salt stress tolerance [28,29]. Consistently, the application of exogenous ABA increases the proline levels and impairs the inhibiting effect of salt stress [14,30]. The rapidly increased endogenous ABA levels then enhance the ABA signaling activates sucrose nonfermenting 1-related protein kinases (SnRK2s) [31], which plays a key role in regulating osmotic homeostasis by activating the BAM1- and AMY3-dependent breakdown of starch into sugar and sugar-derived osmolytes [32].

Tre is a non-reducing disaccharide composed of two glucose molecules with 1,1-glycosidic bonds, and its metabolic precursor is trehalose 6-phosphate (T6P). The main metabolic pathways of T6P are as follows: uridine glucose diphosphate (UDPG) and glucose-6-phosphate (G6P) are catalyzed by trehalose 6-phosphate synthase (TPS) to synthesize T6P. T6P is then dephosphorylated and turns into Tre via trehalose-6-phosphate phosphatase (TPP) [33]. Moreover, Tre metabolism is regulated by a large gene family. Rice (*Oryza sativa* L.) has 11 *TPS* genes and nine *TPP* genes, potentially modulated by the T6P and trehalose levels [33,34]. Studies have shown that Tre and its metabolites related to synthesis, especially T6P, play an important role in regulating plant growth and development and resisting biotic and abiotic stress [34]. In a high yielding rice IR64, the enhancement of Tre has been found to elevate tolerance against drought, salinity, and sodic conditions [35]. Tre improves antioxidant activities and is also involved in signaling association with signaling molecules and phytohormones, improving the plant performance under salt stress [36]. Ge et al. [37] showed that the rice trehalose-6-phosphate phosphatase gene, *OsTPP1*, could be rapidly and transiently induced by salt stress, osmotic stress, and abscisic acid treatment. *OsTPP1* is also regulated by *OsICE1* (Inducer of CBF Expression1), a key transcription factor involved in the regulation of cold tolerance in response to cold stress [38]. The synthesis of Tre in *tpp1* mutant seeds was blocked, resulting in the inhibition of ABA catabolism genes *OsABA8ox2/3* expression, thereby delaying germination [39]. In addition, the *OsTPP7* gene is involved in the regulation of the hypoxic germination of rice seeds. Deletion of *OsTPP7* is one of the reasons for the poor hypoxic germination of many modern rice varieties [40]. The *Arabidopsis AtTPPE* gene is regulated by ABF2, an important transcription factor in the ABA signaling pathway, which is involved in the regulation of root growth and stomatal closure by ABA [41]. The mutation of *OsTPS8* significantly reduced the content of soluble sugars including Tre, and thus decreased the salt tolerance by altering the ABA-regulated suberin deposition [42]. The expression of *OsTPP3* was induced by drought, and the overexpression of *OsTPP3* increased the drought tolerance of rice seedlings. Furthermore, ABA biosynthesis was also induced in the *OsTPP3*-overexpressing lines [43]. However, the relationship between *OsTPP3* and ABA in regulating salt tolerance is still unknown.

In this study, we found that the application of exogenous Tre significantly increased the salt tolerance of rice seedlings by enhancing the antioxidant enzyme activities. Knockout of the *OsNCED3* gene reduced the salt tolerance and Tre content, while the application of ABA and Tre both increased the salt tolerance of the *nced3* mutant. Further study revealed that *OsTPP3* was induced by both ABA and salt stress. The overexpression of *OsTPP3* increased salt tolerance while the knockout of *OsTPP3* decreased the salt tolerance of the rice seedlings. Together, our results demonstrate that salt-enhanced ABA upregulated the expression of *OsTPP3*, resulting in the accumulation of Tre and thus increased salt tolerance.

## 2. Results

### 2.1. Application of Exogenous Tre Promotes Salt Tolerance of Rice Seedlings

Salt stress is one of the leading threats to crop growth and productivity. Tre has been proven to be a promising osmo-protectant against salt stress in many plants [36]. In this study, we found that 8 day old rice seedlings were significantly injured by salt stress (100 mM NaCl) after treatment for 7 days. However, the application of 0.5% Tre solution by spraying the leaves significantly impaired the damage by salinity (Figure 1a,b). Although the dry weight of the rice seedlings under salt stress was not significantly increased by Tre (Figure 1e), the survival rate of the seedlings under salt stress was significantly enhanced by exogenous Tre (Figure 1d). Furthermore, the content of malonaldehyde (MDA) was also significantly reduced by Tre at 7 d of salt treatment (Figure 2a). The antioxidant enzyme assay revealed that the application of Tre increased the activities of catalase (CAT) and superoxide dismutase (SOD) (Figure 2b,c), but slightly reduced the activity of peroxidase (POD) (Figure 2d), suggesting that the application of Tre improved the antioxidant enzymes and thus decreased the oxidant damage of salt stress.

### 2.2. OsNCED3 Was Induced by Salt Stress and Led to the Accumulation of Trehalose

ABA plays a central role in regulating the salt tolerance of rice seedlings [11]. To reveal the relationship between ABA and Tre in seedlings under salt stress, we examined the expression of key genes for ABA biosynthesis. *OsNCED1* was the most abundant ABA biosynthesis gene in the rice seedling and was not induced by salt stress (Figure 3a). However, we found that *OsNCED3* and *OsNCED5*, which were slightly expressed in rice seedlings, were significantly induced by salt stress (Figure 3a). Since *OsNCED3* was critical for ABA biosynthesis in rice seedlings under drought stress [44], we thus generated the *OsNCED3*-overexpressing (OE1) and *nced3* mutant (KO1) lines. As shown in Figure 3, the overexpression of *OsNCED3* significantly enhanced the salt tolerance and survival rate (Figure 4a–d). In contrast, the knockout of *OsNCED3* by CRISPR/Cas9 significantly reduced the salt tolerance and survival rate (Figure 4a–d). The dry weight of both OE1 and KO1 was lower than that of the wide type plant (ZH11) (Figure 4e). Then, the Tre content in OE1 and KO1 was analyzed by using the LC-MS/MS platform. Interestingly, we found that the Tre content was increased by overexpressing *OsNCED3* and decreased in the nced3 mutant (Figure 4f). As the precursor of Tre, the T6P content was significantly decreased by overexpressing *OsNCED3*, but increased in the *nced3* mutant (Figure 4f), suggesting that *OsNCED3*-dependent ABA might play a key role in regulating trehalose biosynthesis. Indeed, the expressions of the ABA biosynthesis genes were not changed by exogenous Tre in the salt-treated seedlings (Figure 3b), indicating that ABA functions upstream of Tre in response to salt stress.

To further reveal the relationship between ABA and Tre in the rice seedlings under salt stress, the salt-treated KO1 mutant seedlings were sprayed with ABA and Tre. The application of ABA restored the ABA level in the KO1 mutant and thus increased the salt tolerance when compared with ZH11 (Figure 5a–c). As expected, spraying the KO1 seedlings with Tre also enhanced the salt tolerance of the KO1 mutant compared with the wild type rice ZH11 (Figure 5a–c). The survival rates of KO1 sprayed with ABA and Tre were slightly higher, but without significance, than that of ZH11 sprayed with H_2_O, indicating that, similar to ABA, the application of Tre was also able to partially recover the salt tolerance of the KO1 mutant.

### 2.3. ABA-Induced OsTPP3 Positively Regulated Salt Tolerance of Rice Seedlings

Tre is catalyzed by TPP from trehalose-6-phosphate in plants. In rice, 11 *OsTPP* genes have been found to encode trehalose-6-phosphate phosphatase [45]. To identify the target for ABA in seedlings under salt stress, qRT-PCR was performed in the rice seedlings under ABA and NaCl treatments for 1 d, 3 d, and 5 d. Eight out of the 11 *OsTPP* genes were detected in our experiments (Figure 6). *OsTPP5*, *OsTPP6*, and *OsTPP7* were not induced or suppressed by both ABA and NaCl. *OsTPP1* was induced by NaCl and inhibited by ABA while *OsTPP8* was induced by ABA but decreased by NaCl. *OsTPP2* and *OsTPP4* were suppressed by both ABA and NaCl treatment. Unlike the above *OsTPP* genes, *OsTPP3* was induced by ABA at 1 d and induced by NaCl at 3 d and 5 d, suggesting that *OsTPP3* might be the target for salt-enhanced ABA in rice seedlings under salt stress.

*OsTPP3* has been reported to positively regulate drought stress. However, little is known about the function of *OsTPP3* in rice seedlings under salt stress. To answer this question, we constructed the *OsTPP3*-overexpressing line (OE10) and *tpp3* mutant line (KO2). Consistently, the overexpression of *OsTPP3* significantly increased the salt tolerance of the rice seedlings (Figure 7a–c); the MDA content in OE10 was significantly decreased at 7 d (Appendix A) and the survival rate of OE10 was significantly higher than that of ZH11 (Figure 7d). In contrast, the *tpp3* mutant was much more sensitive to salt stress (Figure 7a–c), the MDA content in KO2 was significantly enhanced at 5 d and 7 d (Appendix A), and the survival rate of KO2 was much lower than that of the ZH11 seedlings (Figure 7d). The dry weight of OE10, ZH11, and KO2 after salt treatment was similar (Figure 7e), indicating that manipulating the *OsTPP3* gene does not affect the growth of rice seedlings. In addition, similar to the transgenic lines of *OsNCED3* (OE1 and KO1), the Tre content was significantly enhanced in OE10 and decreased in KO2, while the T6P content was decreased in OE10 and enhanced in KO2 (Figure 7f), suggesting that *OsTPP3* plays a key role in the Tre biosynthesis of rice seedlings. Then, the KO2 seedlings were again used to spray with ABA and Tre under the salt stress condition. As can be seen in Figure 8, the application of both ABA and Tre also partially recovered the salt tolerance of the KO2 seedlings. Application of ABA and Tre both increased the survival rate (Figure 8d). These results suggest that the ABA-induced *OsTPP3* positively regulates the salt tolerance of rice seedlings.

## 3. Discussion

### 3.1. Exogenous Tre Enhances the Salt Tolerance of Rice Seedlings by Improving Antioxidant Activities

Trehalose is a non-reducing sugar that is synthesized in plants and plays a protector role in response to different abiotic stresses [46]. One of the most important functions of Tre in the salt stress response of plants is as an osmo-protectant. However, Tre can also stabilize cellular membranes, proteins, and reduce the damaging impacts of salt stress on the biological system of the plant [47]. In addition, recent studies have proven that Tre, like its precursor T6P, is involved in regulating gene expression in different biological processes including genes for plant hormone and antioxidant enzymes [39,48]. In the present study, we also found that the application of a 0.5% Tre solution by spraying the leaves significantly increased the survival rate of the WT seedlings. The dry matter of the Tre-treated seedlings showed a slight increase compared to the control group treated with H_2_O, but there was no significant difference between them (Figure 1). In agreement with the survival rate, the MDA content in the seedlings sprayed with Tre was also significantly decreased compared with that of the control group sprayed with H_2_O (Figure 2a). The antioxidant enzyme activities analysis showed that CAT and SOD were much higher in the seedlings treated with Tre than that of CK (Figure 2b,c). Generally, salt stress increases ROS production and causes cell membrane damage and lipid peroxidation, resulting in increased electrolyte leakage and reduced membrane permeability [49]. It was documented that Tre plays a major role in scavenging O^2−^ by enhancing the activity of SOD [50]. Furthermore, CAT is considered a bulk removal of H_2_O_2_ produced under salt stress, although CAT has a low affinity for H_2_O_2_ [51]. Indeed, Tre increased the expression of different antioxidant genes (Cu/Zn-SOD, Fe-SOD, Mn-SOD, POD, and CAT) that countered the effects of salt stress and enhanced plant performance [36]. On the other hand, ROS also act as an important signaling molecule in response to salt stress and must be maintained at a basal level in plants, known as ROS homeostasis [11]. Together, these studies indicate that a Tre mediated increase in antioxidant activities substantially scavenges the ROS and protects crops from salt-induced oxidative damage.

### 3.2. OsNCED3 Plays a Key Role in Regulating the Accumulation of Tre in Rice Seedlings in Response to Salt Stress

Recently, more and more evidence suggests that Tre is more than an osmo-protector during plant growth and stress responses. We found that the knockout of the *OsTPP1* gene led to a lower level of Tre during seed germination, which resulted in the downregulation of the *OsABA8ox2* and *OsABA8ox3* genes [39], indicating that Tre might act as a signaling molecule in regulating gene expression. Indeed, the application of exogenous Tre was also found to increase the expression of *NCED1* and *NCED2*, two ABA biosynthesis genes, under salt stress [52]. Furthermore, these results also revealed a closed relationship between Tre and ABA. ABA, a well-known stress responsive plant hormone, plays a major and irreplaceable role in salt stress. In response to salinity and osmotic stress, the ABA biosynthesis genes *OsNCED3* and *OsNCED5*, encoding 9-cis-epoxycarotenoid dioxygenases, increased the ABA levels and enhanced salt stress tolerance [28,29]. Here, we also found that the expression of *OsNCED3* and *OsNCED5* was significantly induced by salt stress, especially *OsNCED3*, which was induced over 10 times (Figure 3a), suggesting that *OsNCED3* might be the most important gene for ABA biosynthesis under salt stress. Indeed, the overexpression of *OsNCED3* significantly increased the salt tolerance of rice seedlings. In contrast, the mutant of *OsNCED3* displayed a much higher sensitivity to salt stress than the WT seedlings (Figure 4). The expression pattern of *OsNCED3*, a key gene for ABA biosynthesis, revealed that ABA can be synthesized in leaf vascular parenchymal cells in response to different abiotic stresses [53]. In response to salt stress, ABA-activated SnRK2s also regulate osmotic homeostasis through regulating the BAM1- and AMY3-dependent breakdown of starch into sugar and sugar-derived osmolytes [11]. Interestingly, the metabolite analysis revealed that the Tre content also significantly increased in the *OsNCED3*-overexpressing seedlings and decreased in the *nced3* mutant (Figure 4f). In addition, the salt tolerance of the *nced3* mutant was increased not only by exogenous ABA, but also by the application of Tre (Figure 5), suggesting that *OsNCED3*-dependent ABA enhancement in response to salt stress elevated the biosynthesis of Tre to deal with the damage by salt stress. The use of an ABA inducible promoter has been found to be useful in the case of driving the expression of Tre biosynthesis genes to avoid energy drain under the non-stress condition in plants and to minimize the number of foreign genes [7,35]. In *Arabidopsis*, the expression of *AtTPPE* is induced by ABA treatment, which enhances the accumulation of ROS and thus regulates root growth and stomata movement [41]. Similarly, *AtTPPI* is also regulated by ABA in response to drought stress and positively regulates root growth and stomatal apertures [54]. On the other hand, *OsTPS8*, another key gene for trehalose biosynthesis, is induced by salt stress. Mutation of *OsTPS8* led to a lower level of Tre and other soluble sugars via the expression of genes in the ABA signaling pathway [42]. Furthermore, Tre application improves the accumulation of osmolytes and maintains hormonal crosstalk, which in turn enhances the plant performance under salt stress, indicating that the crosstalk between Tre and ABA is crucial for salt tolerance [36].

### 3.3. OsTPP3 Is Responsible for the ABA-Induced Accumulation of Tre under Salt Stress

Thirteen *OsTPP* genes have been identified in rice, among which *OsTPP1*, *OsTPP2*, and *OsTPP7* have been shown to be active in catalyzing the conversion of T6P to Tre in plants [45]. *OsTPP1* has been reported to be induced by osmotic stress and salt stress [37], which was also found in this study. However, the expression of *OsTPP1* was not changed by the treatment of ABA. In contrast, the expression of *OsTPP8* was rapidly induced by ABA treatment, but not by salt stress (Figure 6). *OsTPP3* was demonstrated to positively regulate the drought tolerance of rice seedlings. The expression of *OsTPP3* was induced by drought stress. OE lines of *OsTPP3* displayed a higher tolerance to drought than the WT seedlings and were more sensitive to ABA treatment, indicating a close relationship between the *OsTPP3* gene and ABA [43]. In the present study, the expression of *OsTPP3* was significantly induced by ABA at 1 day after treatment and increased by salt stress at 3 days and 5 days after treatment (Figure 6). Our results demonstrate that *OsTPP3* was induced by the salt-enhanced ABA in rice and plays a positive role in regulating the salt tolerance of rice seedlings. In a drought resistant plant, *Oropetium thomaeum*, trehalose accumulation was found to be increased with the application of ABA via the upregulation of Tre biosynthesis genes [55]. However, it is worth noting that the expression of *OsTPP1* and *OsTPP3* was reduced in inferior grains of the *aba8ox2* mutant, an ABA catabolism gene, resulting in an elevated T6P level and thus promoting grain filling [56], indicating that an elevated ABA content might not always increase the expression of *OsTPP3* in different tissues of rice. In addition, the overexpression of *TaTPP*-*7A* promoted the grain filling of wheat kernels and induced the expression of genes involved in the ABA signaling pathway [57]. In *Arabidopsis*, the expression of the ABI4 gene, which encodes a key component of the ABA signaling pathway, is regulated by the *AtTPS1* gene [58]. These results suggest that the relationship between ABA and Tre might be more complicated.

## 4. Materials and Methods

### 4.1. Plant Materials and Growth Conditions

For the overexpression line of *OsTPP3* (OE10) and *OsNCED3* (OE1), the full-length coding sequences of *OsTPP3* (LOC_Os07g43160) and *OsNCED3* (LOC_Os03g44380) were cloned and inserted into a vector driven by the Ubi promoter, respectively. The ligation and transformation were performed by the One Step Cloning Kit (Vazyme Biotech Co., Ltd., Nanjing, China). After identification and sequencing, the correct recombinant vectors were used for genetic transformation. The genetic transformation was completed by BioRun Bio (Wuhan, China). For the CRISPR/Cas9 mutant lines of *OsTPP3* (KO2) and *OsNCED3* (KO1), the target sites for KO2 (GCTCACCGACTGCAAGAAGG) and KO1 (ATTCGGTGAGATTCTCCGCG) were designed using the website https://crispr.dbcls.jp/, accessed on 18 January 2021. [59]. The CRISPR/Cas9 vector construction method was performed as previously described [60]. The japonica rice variety Zhonghua 11 (ZH11) was used as the genetic background for all transgenic lines in this study.

Seeds of rice were sterilized with 75% ethanol for 1–2 min, then soaked in a 20% sodium hypochlorite solution for 20 min. Finally, they were rinsed with distilled water 3–5 times until clean. The sterilized rice seeds were soaked in distilled water overnight and then evenly spread in Petri dishes covered with filter paper. Seeds were cultured at 30 °C in an incubator in the dark for 2–3 days until the radicle emergence of >1 mm. The germinated seeds were then placed into a plant hydroponic culture box with Yoshida normal nutrient solution and grown in a constant temperature light incubator under simulated natural light conditions (14-h day/10-h night) with a light strength of 3000 lux at 30 °C/26 °C (day/night) cycles. The nutrient solution was replaced every three days. After being cultured for eight days, the seedlings were further treated.

### 4.2. Salt, Trehalose, and ABA Treatments

For salt treatment, 8-day old seedlings were treated with Yoshida nutrient solution with the addition of 100 mM NaCl, named NaCl (62 Yoshida, 1981), while the control group was treated with the normal Yoshida nutrient solution, named CK. For the treatment with Tre and ABA, 0.5% (*w*/*v*) Tre and 5 μM ABA were sprayed on the leaves of the rice seedlings on the first, third, and fifth day after NaCl treatment, respectively. The control groups were treated with ddH_2_O spray, named H_2_O. The phenotypes were recorded using photography. Seedlings were treated for 7 days, and then subsequently cultured with a normal Yoshida nutrient solution recovery for 7 days. The survival rates (percentage of live seedlings in all tested plants) were measured after 7 days of salt treatment and 7 days of recovery.

### 4.3. RNA Extraction and Real-Time Quantitative RT-PCR (qRT-PCR) Analysis

The total RNA was extracted using the TransZol Up Plus RNA Kit (Transgen, Beijing, China) from the leaves of the rice seedlings, and ≥1 μg of total RNA was reverse transcribed using a HiScript First-Strand cDNA Synthesis Kit (Vazyme, Nanjing, China). To detect the expression levels of genes, qRT-PCR was performed with Hieff qPCR SYBR Green Master Mix (YEASEN, Shanghai, China). Data were standardized to the internal reference gene *Actin1*, and relative quantification was used for data analysis. Three biological replicates were performed in each experiment.

### 4.4. Determination of T6P and Tre Contents

The T6P and Tre contents were detected by the Institute of Crop Science, Chinese Academy of Agricultural Sciences. Samples were frozen in liquid nitrogen and ground into powder. About 0.1 g of the sample powder was weighed and its content determined using the liquid chromatography tandem mass spectrometry (LC-MS/MS) method. Three replicates were measured for each experiment and each replicate included five rice seedlings.

### 4.5. Determination of Dry Matter Weight

Samples of the above-ground parts of the rice seedlings were collected and dried in an oven at 105 °C until a constant weight was achieved. Then, the dry matter weight of the plant samples was weighed and recorded. Each treatment had five biological replicates, with each replicate consisting of five rice seedlings.

### 4.6. Enzyme Activity Analyses

To detect the enzyme activities including MDA, CAT, SOD, and POD, the corresponding Solarbio Assay Kits (Beijing Solarbio Science & Technology Co., Ltd., Beijing, China) were used in this study. Samples of the fresh rice seedling leaves were collected and rapidly frozen in liquid nitrogen, then 0.1 g of the sample was weighed and homogenized with 1 mL extraction solution in an ice bath. Then, it was centrifuged at 8000× *g* at 4 °C for 10 min. The supernatant was utilized for the determination of the enzyme activities.

For the activity of malondialdehyde (MDA), 200 μL of the supernatant was mixed with MDA reagents provided in the assay kit and heated at 100 °C for 60 min in a water bath, followed by cooling in an ice bath. Then, the samples were centrifuged at 10,000× *g* for 10 min at room temperature. The supernatant was measured for absorbance at 532 nm and 600 nm.

For the activity of catalase (CAT), 10 μL of the supernatant was added to the pre-treated 190 μL CAT working solution provided by the assay kit and mixed for 5 s. The absorbances were measured at 240 nm instantly and after 1 min at room temperature, respectively.

For the activity of superoxide dismutase (SOD), 20 μL of the supernatant was mixed with reagents provided in the assay kit and heated at 37 °C for 30 min in a water bath. The absorbance was measured at 560 nm.

For the activity of peroxidase (POD), 5 μL of the supernatant was added to the pre-treated reagents provided in the assay kit, following the instructions in the manual. The absorbances were measured at 470 nm after 30 s and 1 min and 30 s, respectively.

### 4.7. Statistical Analysis

Data processing and analysis were performed using Microsoft Excel 2019 and GraphPad Prism software (GraphPad Software Inc., La Jolla, CA, USA). The significant difference was analyzed using the Student’s *t* test and one-way ANOVA. * *p* < 0.5 represents a significant difference, ** *p* < 0.01 represents an extremely significant difference.

## 5. Conclusions

Taken together, our study demonstrates that ABA accumulation via the upregulation of *OsNCED3* activates the expression of *OsTPP3*, resulting in the elevation of Tre content and thus the salt tolerance of the rice seedlings. The application of exogenous Tre increased the salt tolerance of rice seedlings by activating antioxidant enzymes. Salt stress significantly induced the expression of *OsNCED3* and ABA content, which was found to be crucial in the upregulation of the *OsTPP3* gene in rice seedlings under salt stress. The upregulation of *OsTPP3* thus promoted the accumulation of Tre and eventually increased the salt tolerance of the rice seedlings.

## Figures and Tables

**Figure 1 plants-12-02665-f001:**
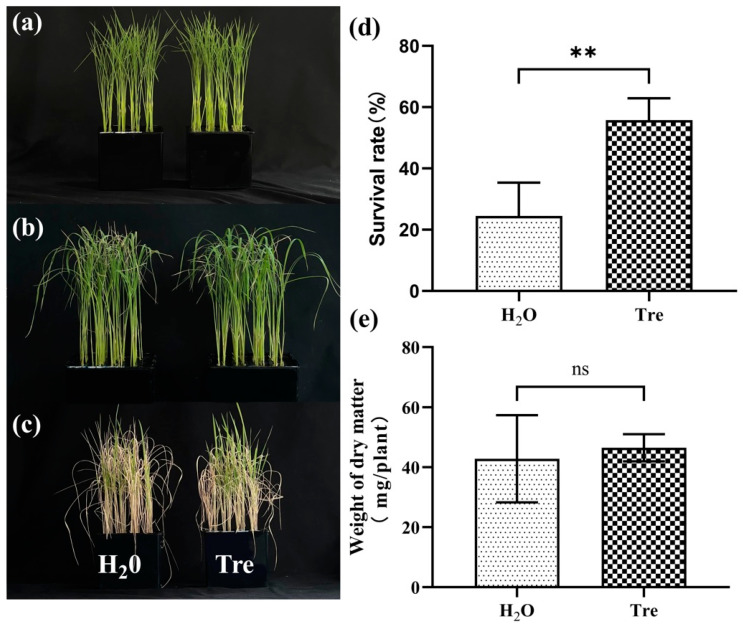
**Application of** Tre **enhanced salt tolerance of rice seedlings.** (**a**) The 8-day old seedlings of wild type rice (ZH11) before salt stress treatment. (**b**) Phenotype of rice seedlings after 7 days of salt treatment with or without 0.5% Tre sprayed. H_2_O was the control group that was sprayed with ddH_2_O at 1 d, 3 d, and 5 d after salt stress. Tre was the treatment group that was sprayed with 0.5% Tre at 1 d, 3 d, and 5 d after salt stress. (**c**) Phenotype of seedlings recovered for 7 days after salt treatment. (**d**) The survival rate was measured after 7 days of salt treatment with or without 0.5% Tre sprayed and 7 days of recovery. (**e**) The dry matter weight was measured after 7 days of salt treatment with or without 0.5% Tre sprayed and 7 days of recovery. The vertical bar indicates the means of three replicates ± SD, each replicate included 48 rice seedlings. (** *p* < 0.01, ns means not significant, one-way ANOVA).

**Figure 2 plants-12-02665-f002:**
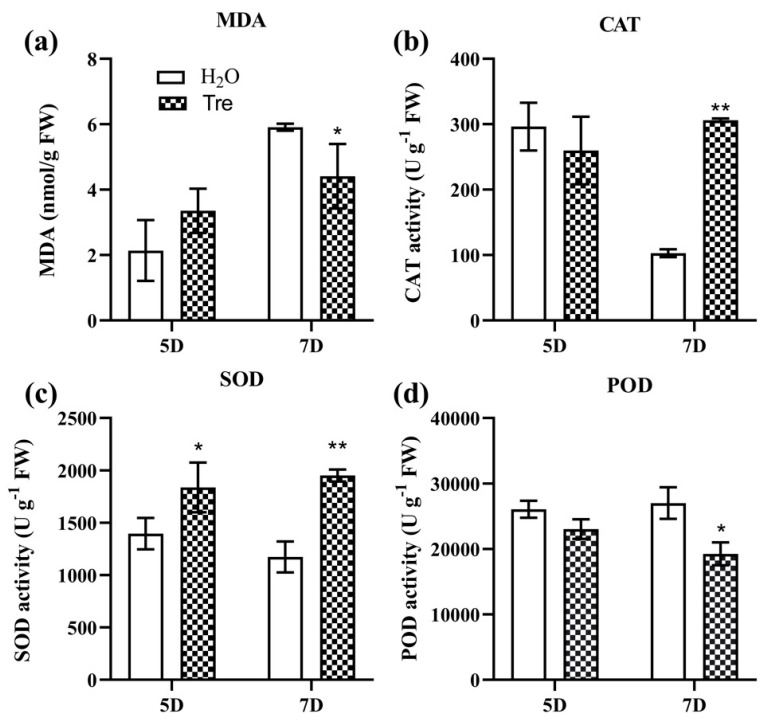
**Analysis of the MDA and antioxidant enzyme activities.** MDA content (**a**), CAT activity (**b**), SOD activity (**c**), and POD activity (**d**) in the leaves of the salt-treated seedlings sprayed with H_2_O and Tre, respectively. The vertical bar indicates the mean of three replicates ± SD. One-way analysis of variance was used to conduct the comparisons, * *p* < 0.05, ** *p* < 0.01 was considered as statistically significant, ns means not significant.

**Figure 3 plants-12-02665-f003:**
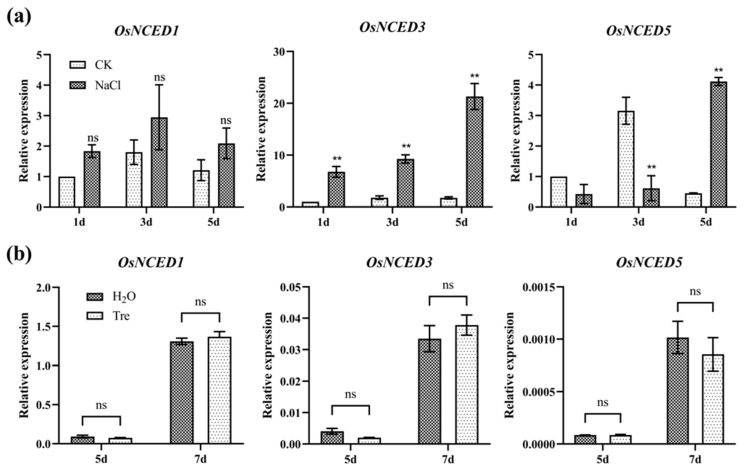
**Expression analyses of key genes for ABA biosynthesis in the salt-treated seedlings.** (**a**) Expression of *OsNCED1*, *OsNCED3*, and *OsNCED5* in the seedlings treated with salt for 1 d, 3 d, and 5 d. (**b**) Expression of *OsNCED1*, *OsNCED3*, and *OsNCED5* in the salt-treated seedlings sprayed with H_2_O and Tre, respectively. The vertical bar indicates the mean of three replicates ± SD. One-way analysis of variance was used to conduct the comparisons, ** *p* < 0.01 was considered as statistically significant, ns means not significant.

**Figure 4 plants-12-02665-f004:**
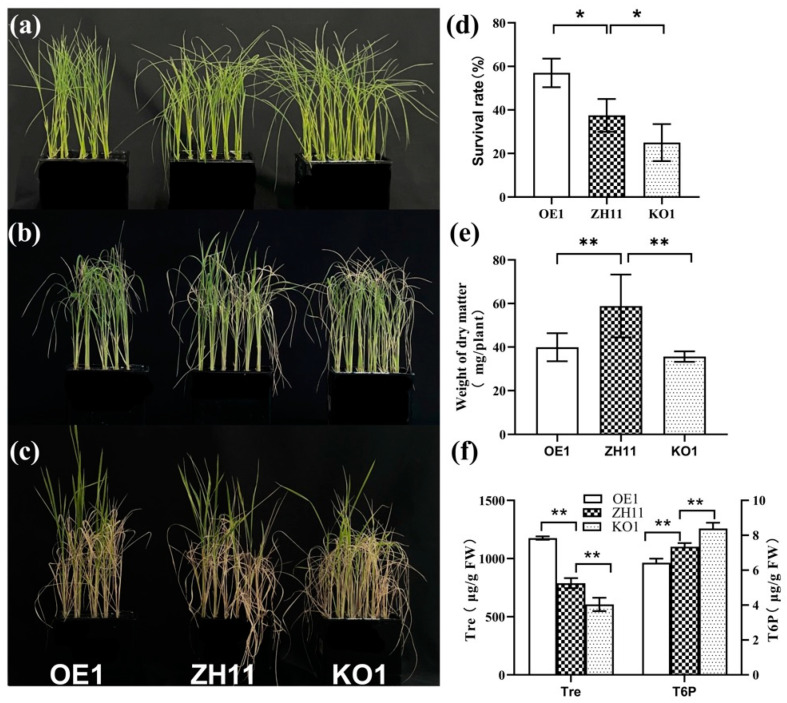
***OsNCED3* positively regulated the salt tolerance and** Tre **accumulation of rice seedlings.** (**a**) The 8-day old seedlings of the *OsNCED3* overexpressed line (OE1), wild type (ZH11), and *nced1* mutant (KO1) before salt stress treatment. (**b**) Phenotype of the rice seedlings 7 d after salt stress. (**c**) Phenotype of seedlings recovered for 7 days after salt treatment. (**d**,**e**) Survival rate and dry weight of the salt treated seedlings of OE1, ZH11, and KO1. (**f**) The Tre and T6P content of the salt-treated seedlings for 7 d. The vertical bar indicates the mean of three replicates ± SD, each replicated included 48 rice seedlings. One-way analysis of variance was used to conduct the comparisons, * *p* < 0.05 or ** *p* < 0.01 was considered as statistically significant.

**Figure 5 plants-12-02665-f005:**
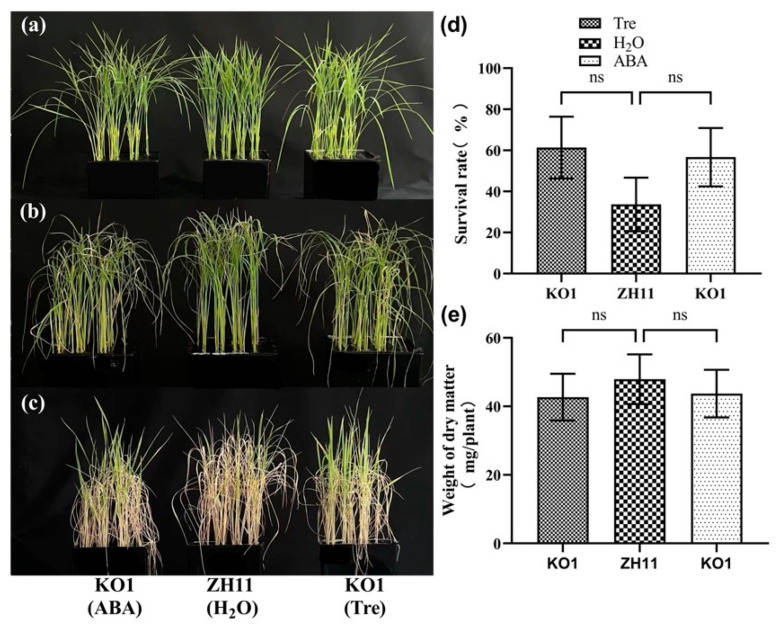
**Application of ABA and Tre significantly enhanced the salt tolerance of the *nced1* mutant.** (**a**) The 8-day old seedlings of ZH11 and KO1 before salt stress treatment. (**b**) Phenotype of the rice seedlings 7 d after salt stress, ABA, H_2_O, and Tre mean seedlings were sprayed with ABA, H_2_O, and 0.5% Tre at 1 d and 3 d after salt stress. (**c**) Phenotype of the seedlings recovered for 7 days after salt treatment. (**d**,**e**) Survival rate and dry weight of the salt treated seedlings of ZH11 and KO1. The vertical bar indicates the mean of three replicates ± SD, each replicated included 48 rice seedlings. One-way analysis of variance was used to conduct the comparisons, ns means not significant.

**Figure 6 plants-12-02665-f006:**
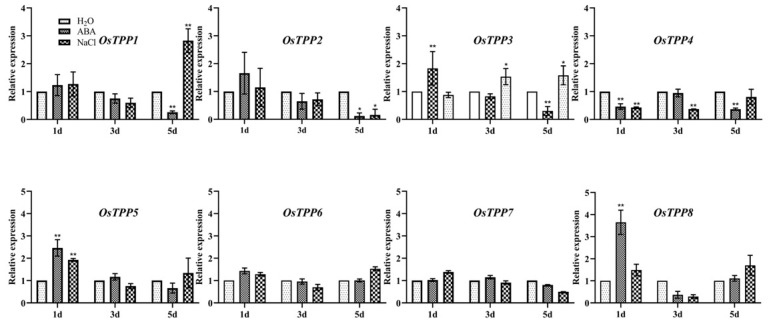
**Expression analyses of the *OsTPP* gene families in seedlings under ABA and salt treatments.** The 8-day old seedlings were treated with 5 μM ABA and 100 mM NaCl, respectively, and sampled at 1 d, 3 d, and 5 d for the expression analysis of *OsTPP* genes. The vertical bar indicates the mean of three replicated ± SD. One-way analysis of variance was used to conduct the comparisons, * *p* < 0.05 or ** *p* < 0.01 was considered as statistically significant.

**Figure 7 plants-12-02665-f007:**
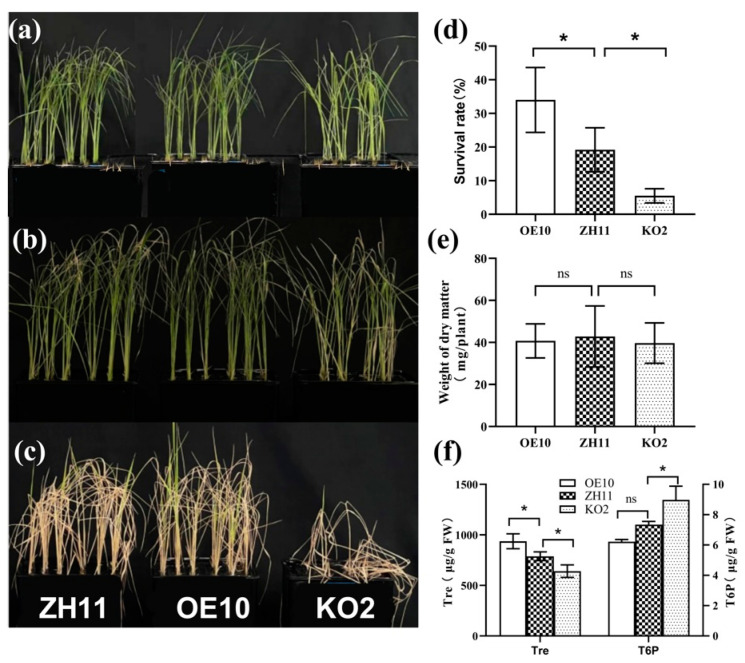
***OsTPP3* positively regulated the salt tolerance and** Tre **accumulation of the rice seedlings.** (**a**) The 8-day old seedlings of the *OsTPP3* overexpressed line (OE10), wild type (ZH11) and *tpp3* mutant (KO2) before salt stress treatment. (**b**) Phenotype of rice seedlings 7 d after salt stress. (**c**) Phenotype of seedlings recovered for 7 days after salt treatment. (**d**,**e**) Survival rate and Dry weight of salt treated seedlings of OE10, ZH11 and KO2. (**f**) Tre and T6P content of salt-treated seedlings for 7 d. Vertical bar indicates the mean of three replicated ±SD, each replicate included 48 rice seedlings. One-way analysis of variance was used to conduct the comparisons, * *p* < 0.05 was considered as statistically significant, ns means not significant.

**Figure 8 plants-12-02665-f008:**
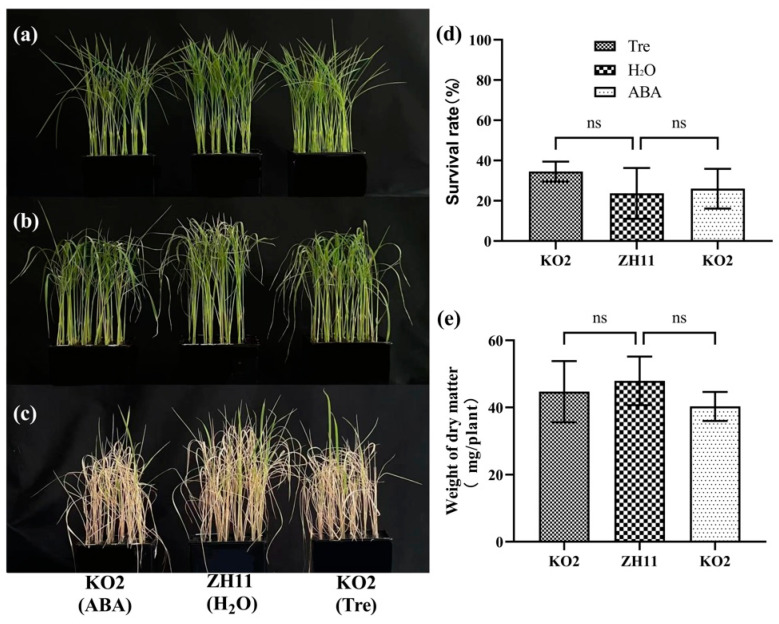
**Application of Tre enhanced the salt tolerance of the *tpp3* mutant.** (**a**) The 8-day old seedlings of ZH11 and KO2 before salt stress treatment. (**b**) Phenotype of the rice seedlings after 7 days of salt treatment with ABA, H_2_O, and 0.5% Tre sprayed. Spraying method: ABA, H_2_O, or 0.5% Tre sprayed at 1 d, 3 d, and 5 d after salt treatment. H_2_O was the control group. (**c**) Phenotype of the seedlings recovered for 7 days after salt treatment. (**d**,**e**) The survival rate and dry weight of the ZH11 and KO2 rice seedlings were determined after salt treatment. The vertical bar indicates the mean of five replicates ± SD, each replicate included 48 rice seedlings. One-way analysis of variance was used to conduct the comparisons, ns means not significant.

## Data Availability

Not applicable.

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
