# Peer review of "Abscisic Acid Enhances Trehalose Content via *OsTPP3* to Improve Salt Tolerance in Rice Seedlings"

_plants, 2023, doi:10.3390/plants12142665_

Round 1
Reviewer 1 Report
The manuscript ‘Abscisic acid activates OsTPP3 to improve salt tolerance in rice seedlings’ reports research on the role of trehalose in salt stress response in rice.
The topic is interesting, but I found several aspects that need to be addressed before publication.
The work is presented in an inaccurate manner and the discussion is not exhaustive, considering the amount of research available on this topic.
First of all, the title is not adequate since it points to ABA and a specific gene, but the manuscript is more oriented on the role of trehalose. The cross-talk between stress perception, ABA, trehalose, and gene expression is missing. The last figure (Fig. 9) is a very simple representation that does not consider the complexity of the signal transduction pathway involved in stress response. So, I would suggest integrating it with more information, also considering the work described in the manuscript, in which for example gene expression was evaluated for a higher number of genes,
Remaining on figure, Figure 6 is too small, difficult to read, please enlarge it.
In the introduction, there are several missing explanations about gene names and functions, in particular in lines 92-100. The authors could describe more clearly the gene OsTPP(n) family and why they decided to concentrate on OsTPP3 in particular.
In general, check that each abbreviation is defined at first appearance, some examples are specified below (not exhaustive):
Line 92 OsTTP1
Line 127-128 CAT, SOD, POD
Line 156 ZH11 explain that is the wild type
Line 259 CK
In all the experiments it is not clear how many biological replicates have been analysed.
For example, in calculating dry matter, which is expressed as mg/plant, how many plants were utilized? Similarly for the survival rate.
In the captions it is indicate n=3, but I doubt that a survival rate calculated on 3 plants is reliable.
The discussion could be improved since there are many data avalaible in the literature about the specific role of ABA and trehalose in the salt stress response that have not been considered, also regarding other plant species.
Line 287-290 the phrase is not clear
Line 313-314, the phrase is not clear
Line 319-331 This part of the discussion needs a complete rephrasing since there are some mistakes and the meaning is not clear at all.
Materials and methods
The mutant lines should be described more accurately. Were they produced within this research or were they already available? Where previously described in another research? Please specify.
The wild-type lines were ZH11 and CK, this last one is not cited in paragraph 4.1
4.3 the method used for the calculation of the relative expression is not described
4.4 the method used for the quantification of T6P and Tre is not described
Conclusion
Figure 9 should be improved as already said and the caption should be modified accordingly. The caption simply repeats the last sentence of the conclusion (lines 400-405).
The English form is good, but there are some spelling mistakes, for example, slat, instead of salt, in lines 61, 115, 267, 319.
Reviewer 2 Report
The article “Abscisic acid activates OsTPP3 to improve salt tolerance in rice seedlings” by Nenghui et al., represents a great effort in demonstrating the effect of the exogenous application of ABA and trehalose to counteract the effect of stress caused by NaCl in rice, and that has been widely demonstrated in other plant species.
However, some topics must be clarified and addressed before its acceptance:
ABSTRACT
P1; L20: Change “overexpression” to “Overexpression”
1. Introduction
P2; L61: Correct “slat” by “salt”
P3; L115: Correct “slat” by “salt”
2. Results
P5; Figure 3: In figure 3 (a), the authors use the CK code for the control and in 3 (b) they use H2O, regardless of the treatment in both cases it is the control. Standardize the code used to refer to the control.
P8; Figure 6: For a better appreciation of the expression levels, use the same range on the Y-axis scale (for example, 0-4) for the TPP1-TPP4 genes and 0-15 for TPP5-TPP8.
3. Discussion
P9; L258-259: Authors should reconsider the following sentence “although the dry matter Tre-treated seedlings 258 was changed when compared with CK (Fig. 1)”. Since according to the values and the statistical analysis there is no significant difference.
P9; L259-261: The authors refer to CK as the control, but in both figure 1 and 2, the control is referred to as H20. Standardize the code used to refer to the control.
P10; L267: Correct “slat” by “salt”
P11; L318-319: I think the authors in the following sentence “at 2 days and 3 days after treatment” they meant “at 3 days and 5 days after treatment”.
P11; L319: Correct “slat” by “salt”
4. Materials and Methods
P11; L333: The authors do not mention the methodology used to generate the overexpressed lines or knockouts. The promoter used is not described either.
P11; L334 and L336: Correct CRSPRE/Cas9 by CRISPR/Cas9.
P11; L363: Include the methodology used for the extraction and quantification of trehalose instead of the site where the determination was made.
I consider that the quality of the writing is good and only some errors that are pointed out in the review should be addressed.
Round 2
Reviewer 1 Report
The authors took into consideration the suggestions and improved the manuscript accordingly.
There are still some minor issues regarding the abbreviations, the most evident in Tre for trehalose, which is introduced at the first mention and then is not used all the time. The function of the abbreviation is that after the first mention then you use it whenever required, otherwise it is not useful.
Fig. 9 remained the same, and I think it is not useful in this way, my suggestion is to delete it. The scheme is so simple that the figure is not necessary.
Author Response
There are still some minor issues regarding the abbreviations, the most evident in Tre for trehalose, which is introduced at the first mention and then is not used all the time. The function of the abbreviation is that after the first mention then you use it whenever required, otherwise it is not useful.
Response: We are really grateful for the great comments. We have revised the manuscript accordingly.
Fig. 9 remained the same, and I think it is not useful in this way, my suggestion is to delete it. The scheme is so simple that the figure is not necessary.
Response: Many thanks for the comment. We agree with the reviewer and delete the figure 9 in the new manuscript.